# Identification of PNG kinase substrates uncovers interactions with the translational repressor TRAL in the oocyte-to-embryo transition

Masatoshi Hara[1][†], Sebastian Lourido[1], Boryana Petrova[1], Hua Jane Lou[2], Jessica R Von Stetina[1], Helena Kashevsky[1], Benjamin E Turk[2], Terry L Orr-Weaver[1,3]*

[1]Whitehead Institute, Cambridge, United States; [2]Department of Pharmacology, Yale School of Medicine, New Haven, United States; [3]Department of Biology, Massachusetts Institute of Technology, Cambridge, United States

**Abstract** The Drosophila Pan Gu (PNG) kinase complex regulates hundreds of maternal mRNAs that become translationally repressed or activated as the oocyte transitions to an embryo. In a previous paper (Hara et al., 2017), we demonstrated PNG activity is under tight developmental control and restricted to this transition. Here, examination of PNG specificity showed it to be a Thr-kinase yet lacking a clear phosphorylation site consensus sequence. An unbiased biochemical screen for PNG substrates identified the conserved translational repressor Trailer Hitch (TRAL). Phosphomimetic mutation of the PNG phospho-sites in TRAL reduced its ability to inhibit translation in vitro. In vivo, mutation of *tral* dominantly suppressed *png* mutants and restored Cyclin B protein levels. The repressor Pumilio (PUM) has the same relationship with PNG, and we also show that PUM is a PNG substrate. Furthermore, PNG can phosphorylate BICC and ME31B, repressors that bind TRAL in cytoplasmic RNPs. Therefore, PNG likely promotes translation at the oocyte-to-embryo transition by phosphorylating and inactivating translational repressors.
DOI: https://doi.org/10.7554/eLife.33150.001

*For correspondence:
weaver@wi.mit.edu

Present address: [†]Graduate School of Frontier Biosciences, Osaka University, Suita, Japan

Competing interests: The authors declare that no competing interests exist.

## Introduction

One of the most dramatic events in development is the transition from differentiated oocyte to toti-potent embryo, a transition that in nearly all animals occurs in the absence of transcription (*Tadros and Lipshitz, 2009*). Thus, translational control of stockpiles of maternal mRNAs is crucial as the oocyte completes meiosis and resets for embryogenesis, a series of events termed egg activa-tion (*Von Stetina and Orr-Weaver, 2011*). In Drosophila, profound changes in mRNA translation accompany egg activation, with hundreds of maternal mRNAs becoming repressed and nearly a thousand translationally activated (*Kronja et al., 2014*). These translation changes occur in a brief window of less than an hour, and the majority are controlled by the PNG kinase complex (*Kronja et al., 2014*). This kinase complex is composed of the PNG catalytic subunit, whose activity requires the physical association of two activating subunits, GNU and PLU (*Freeman et al., 1986*; *Elfring et al., 1997*; *Fenger et al., 2000*; *Lee et al., 2003*). We recently demonstrated that the activity of PNG is restricted to the window of egg activation by exquisite developmental control of the binding of GNU to PNG and PLU (*Hara et al., 2017*).

PNG is likely to have many targets, given that it controls both mRNAs that become repressed and those that become activated at the oocyte-to-embryo transition (*Kronja et al., 2014*). PNG pro-motes the translation of *smg* mRNA, a translational repressor that can promote deadenylation

(*Tadros et al., 2007*; *Eichhorn et al., 2016*). Most, but not all, of the mRNAs whose translational repression is dependent on PNG undergo SMG-dependent deadenylation (*Eichhorn et al., 2016*). Thus, the role of PNG in translation repression can largely be explained by its effect in activating translation of *smg* mRNA. The mechanisms by which PNG promotes translation of activated mRNAs remain to be uncovered. To determine whether PNG directly controls translational regulators through phosphorylation, we carried out an unbiased biochemical screen to identify PNG substrates. Here, we present the results of that screen and evidence that PNG phosphorylates and inactivates translational repressors.

## Results and discussion

### PNG is a threonine kinase

As an initial approach to identify substrates for the PNG kinase, predicted to be a Ser/Thr kinase, we sought to determine whether PNG phosphorylation occurs at consensus sequences. A positional scanning peptide library (*Mok et al., 2010*) was treated with active PNG kinase complex or a complex with catalytically inactive PNG (KD: kinase dead) purified from Sf9 cells. Peptides were robustly phosphorylated by the active PNG kinase complex in contrast to the kinase-dead control (*Figure 1A*). PNG exhibited a strong preference to phosphorylate threonine, because peptides whose phospho-acceptor site (position 0) was fixed with threonine were strongly phosphorylated, whereas serine peptides were phosphorylated at reduced levels (*Figure 1A,B*). Although no strong consensus sequence was identified, PNG was most strongly selective for hydrophobic amino acids at −3 relative to the phosphorylated residue, and it had some preferences for aromatic residues at position −2 and for arginine at position +2 (*Figure 1B* and *Figure 1—figure supplement 1*). Increased phosphorylation of peptides with threonine present outside of the intended phospho-acceptor position was likely an artifact resulting from the presence of two potential phosphorylation sites.

Kinases with a preference for threonine over serine are atypical, and this specificity is conferred by a beta-branched amino acid residue immediately downstream of the conserved DFG sequence in the kinase activation loop (*Chen et al., 2014*). In PNG, the corresponding amino acid is an isoleucine, which would be predicted to produce a threonine preference (*Figure 1C*).

### Identification of PNG substrates

The peptide arrays did not yield a consensus sequence for PNG of sufficient specificity to be used to identify putative substrates. We previously had identified a limited number of substrates by DIVEC screening, in vitro transcribing and translating Drosophila cDNAs, adding recombinant PNG, and scoring for phosphorylation by gel mobility shift (*Lee et al., 2005*). Because of the limitations of this approach, we designed an unbiased biochemical screen. First, we attempted to introduce a mutation into the gatekeeper residue in the ATP-binding pocket of PNG kinase. Replacing the gatekeeper residue, which is a bulky residue, with a small amino acid allows kinases to utilize ATP analogs to label substrates (*Bishop et al., 2000*; *Alaimo et al., 2001*). Unfortunately, the desired PNG mutants were inactive (*Figure 2—figure supplement 1*).

The alternative strategy we employed to isolate PNG substrates was to use purified recombinant PNG kinase to thio-phosphorylate substrates in embryonic extracts, identifying them by recovery of thio-phosphorylated peptides by mass spectrometry (*Figure 2A*). The endogenous kinases in the extracts from early embryos were inactivated by treatment with 5'-(4-fluorosulphonylbenzoyl)adenosine (FSBA), which covalently binds to kinases at a conserved lysine in the ATP hydrolysis site (*Knight et al., 2012*) (*Figure 2—figure supplement 2*). Wild-type or kinase-dead PNG complex was expressed in Sf9 cells, purified, and added to the extracts with ATP-γS. Western blot analysis with an antibody against alkylated-thio-phosphate (*Allen et al., 2007*) showed that endogenous kinases in the extract had been inactivated, and phosphorylation occurred with wild-type PNG but not the kinase-dead form (*Figure 2B*). Thio-phosphorylated peptides were recovered on iodoacetyl agarose and identified by mass spectrometry (MS) (*Blethrow et al., 2008*; *Rothenberg et al., 2016*). To call a protein a PNG substrate we demanded that at least two independent phosphopeptides were

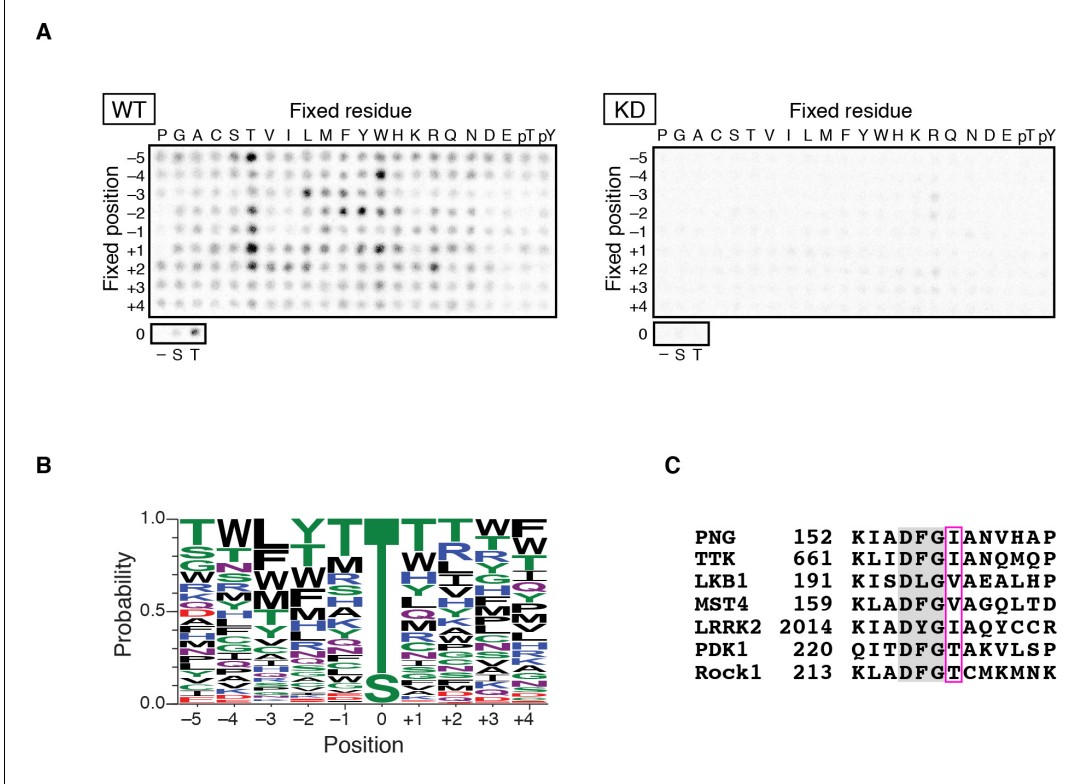

**Figure 1.** PAN GU (PNG) kinase is a threonine-specific kinase. (**A, B**) PNG kinase prefers threonine as a phosphoacceptor site. The peptide library was phosphorylated with PNG kinase wild type (WT) or kinase dead (KD) using radiolabeled ATP (**A**). The signals were quantified and visualized with WebLogo 3.0. (**B**) Ten thousand peptide sequences were generated according to the probabilities predicted from the quantified peptide library data. The colors designate classes of amino acids. (**C**) Alignment of the amino acid sequences near the DFG motif of known threonine-specific kinases and PNG. PNG has a beta-branched residue, isoleucine, immediately downstream of the DFG motif as do other threonine-selective kinases (boxed with magenta (*Chen et al., 2014*). The peptide library screen with WT was repeated in four replicates.

DOI: https://doi.org/10.7554/eLife.33150.002

The following source data and figure supplement are available for figure 1:

**Source data 1.** Quantification of peptide phosphorylation in *Figure 1B*.
DOI: https://doi.org/10.7554/eLife.33150.004

**Figure supplement 1.** Heat map for quantified selective values of PNG phosphorylation.
DOI: https://doi.org/10.7554/eLife.33150.003

recovered. A pilot screen was done with wild-type PNG kinase and 45 proteins were phosphory-lated. A second screen was done in which extracts were treated in parallel with wild-type and kinase-dead PNG. In this second screen, the total representation of peptides in the extract was quantified by doing mass spec analysis of the peptides that did not bind to iodoacetyl agarose. In the second experiment, 36 proteins had at least two independent peptides phosphorylated by wild-type but not kinase-dead PNG. These included 27 of the proteins identified in the pilot experiment (*Figure 2—source data 1*).

A high representation of phosphopeptides was recovered for the translational repressor Trailer Hitch (TRAL) with wild-type but not kinase-dead PNG (*Figure 2C*). Other phosphorylated proteins were ribosomal proteins and translation factors, as well as the PLU activating subunit of the PNG complex. Out of 36 substrates identified, 19 were proteins known to be involved in mRNA transla-tion. Note that the recovery of substrates was not due solely to the abundance of the proteins in the extracts (*Figure 2—source data 1*).

79% of the identified unique peptides had threonine as the phospho-acceptor residue (*Figure 2D*). The threonine preference is consistent with the scanning peptide library result (*Figure 1B*). The identified peptides showed an enrichment of hydrophobic residues at −3 position

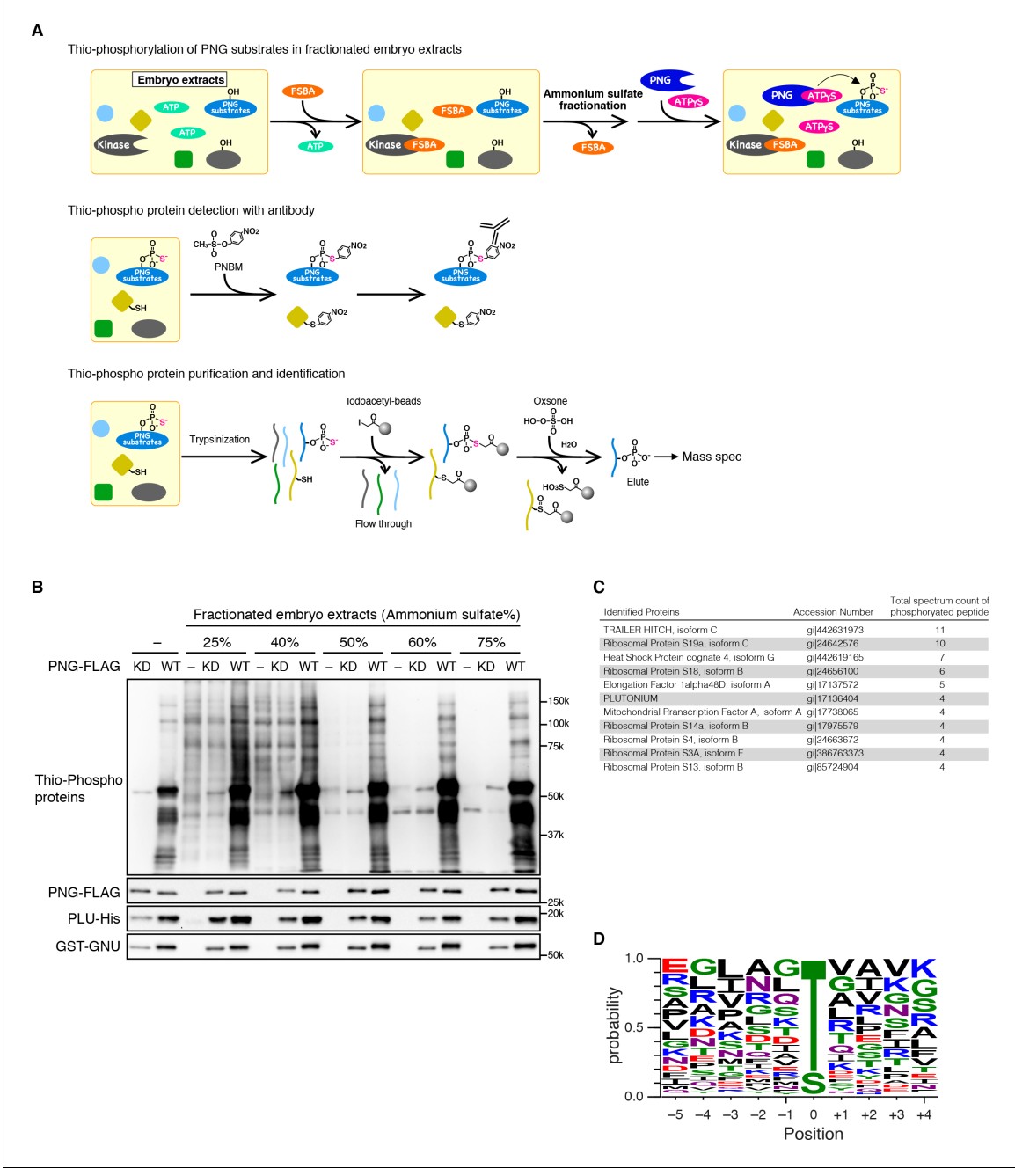

**Figure 2.** Identification of PNG kinase substrates with a biochemical screen. (**A**) A schematic representation of the substrate screen. Embryo extracts were gel-filtrated to exchange the buffer and treated with 5'-(4-Fluorosulfonylbenzoyl)adenosine (FSBA) to inactivate endogenous kinases, followed ammonium sulfate fractionation. The fractionated extracts were dialyzed to remove ammonium sulfate. The recombinant active PNG kinase complex was added to the FSBA-treated fractions with ATP-γS to thio-phosphorylate PNG kinase substrates. To examine thio-phosphorylated proteins by immunoblot, they were alkylated with p-Nitrobenzyl mesylate (PNBM), and the alkylated thiophosphate was detected by its specific antibody. For identification of the thio-phosphorylated peptides, the PNG kinase-treated fractions were pooled and digested with trypsin. Thio-phosphorylated peptides were purified specifically with iodoacetyl beads and oxsone and identified by mass spectrometry. (**B**) Detection of thio-phosphoproteins in the fractions by immunoblots. The FSBA-treated fractions were incubated with wild-type (WT) or kinase-dead (KD) PNG kinase complexes. Thio-phosphorylated proteins were detected as shown in (**A**). The PNG kinase complexes added into the fraction were examined by immunoblots using anti-FLAG (PNG-FLAG), anti-PLU (PLU-His) and anti-GST (GST-GNU) antibodies. (**C**) A list of PNG substrates identified in the second experiment that were recovered with wild-type but not kinase-dead PNG. The proteins found by mass spectrometry following purification of the thio-phosphorylated peptides are listed in the order of the number of total spectrum count of phosphorylated peptides from the sample treated with the wild-type PNG kinase complex. The top 11 of the 36 substrates identified are shown. (**D**) Phosphorylation sites analysis. Peptides identified with >95% probability

*Figure 2 continued on next page*

*Figure 2 continued*

according to Scaffold were used for thio-phosphorylation motif analysis. Phosphopeptides identified in the samples treated with the PNG WT kinase complex but absent from the samples treated with KD kinase were further analyzed. Since elution of the peptides was performed under oxidizing conditions, peptides that differed only in the oxidation state of their methionines were regarded as equal. A total of 112 unique phosphopeptides belonging to 70 different proteins were considered. Using these phosphopeptides, a list of motifs was constructed centered on the phosphosite and including the 5 N-terminal and 4 C-terminal residues found adjacent to the phosphosite on the protein and analyzed with WebLogo 3.0. The screen with the WT kinase was done as a pilot screen, and a second experiment was done in which the results were compared with kinase-dead PNG.

DOI: https://doi.org/10.7554/eLife.33150.005

The following source data and figure supplements are available for figure 2:

Source data 1. Lists of proteins and peptides identified by mass spectrometry analyses.

DOI: https://doi.org/10.7554/eLife.33150.008

Figure supplement 1. Mutation of the gatekeeper residue of PNG inactivates its kinase activity.

DOI: https://doi.org/10.7554/eLife.33150.006

Figure supplement 2. 5'-(4-Fluorosulfonylbenzoyl)adenosine (FSBA) inactivates endogenous kinases in the embryo extracts.

DOI: https://doi.org/10.7554/eLife.33150.007

as in the peptide library, confirming that PNG tends to phosphorylate threonine three residues downstream of a hydrophobic amino acid (*Figure 2D*). The threonine preference was also highly significant (log-odds value of 80.5) in the context of the Drosophila proteome (*O'Shea et al., 2013*). The correspondence with the peptide sequence preference of PNG is further confirmation that the observed phosphopeptides likely reflect direct phosphorylation by PNG. Although the substrates don't reveal a strong PNG consensus sequence, it is possible that interaction between substrates and the PLU or GNU activating subunits may provide specificity beyond that at the phosphorylation site.

## Phosphorylation of TRAL by PNG in vitro

We focused on TRAL, because although there were many more abundant proteins in the extracts, we recovered a high number of PNG-phosphorylated peptides for TRAL. TRAL is a member of the (L)Sm protein family composed of RAP55 in vertebrates, CAR1 in *C. elegans*, and Sdc6 in yeast (*Wilhelm et al., 2005*; *Marnef et al., 2009*). We tested whether PNG can phosphorylate TRAL in vitro. A powerful aspect of the thio-phosphate substrate screen is that the MS analysis identifies the phosphorylated amino acids. 15 amino acids (13 of them threonine), clustered in the C-terminal half of the protein, were phosphorylated by PNG in embryonic extracts (*Figure 3A*). MBP fusions of purified full length TRAL, or the N- and C- terminal fragments were incubated with purified PNG and [γ $^{32}$P]-ATP and analyzed by autoradiography. The full-length protein and the C-terminal half, but not the N-terminal half, were phosphorylated by PNG in vitro (*Figure 3B*). To determine whether PNG-dependent phosphorylation required the amino acids identified in the substrate screens, all 15 were changed to alanine. For both the full-length protein and the C-terminal half, the level of phosphorylation by PNG was reduced with the alanine-substituted forms (*Figure 3B*). Residual phosphorylation of the alanine-substituted form of TRAL raises the possibility that there are other potential PNG phosphorylation sites in the C-terminus of TRAL that were not detected in the screen.

We next wanted to investigate whether phosphorylation of TRAL by PNG inhibits its activity. RAP55 from Xenopus and Sdc6 from yeast are able to inhibit translation in vitro (*Tanaka et al., 2006*; *Nissan et al., 2010*), in yeast apparently by blocking the function of the eIF4G subunit of the eIF4F initiation factor (*Rajyaguru et al., 2012*). We examined translation of an mRNA encoding Myc-tagged GFP in reticulocyte lysates and found that as for other family members, addition of Drosophila TRAL inhibited translation (*Figure 3C*). Because in the in vitro reaction purified PNG does not phosphorylate TRAL to full stoichiometry, we evaluated the effect of PNG phosphorylation by generating a phosphomimetic form of TRAL in which aspartic acid was substituted for the fifteen PNG phosphorylation sites. Strikingly, the phosphomimetic mutations suppressed the translational repression by TRAL (*Figure 3C*). The potential existence of additional PNG phosphorylation sites in the C-terminus of TRAL could account for why suppression of translational repression by the phosphomimetic form of TRAL was not complete. In contrast, TRAL in which these residues were replaced by alanine still inhibited translation of the reporter mRNA in the extracts (*Figure 3C*). These results are consistent with phosphorylation of TRAL by PNG relieving its ability to repress translation.

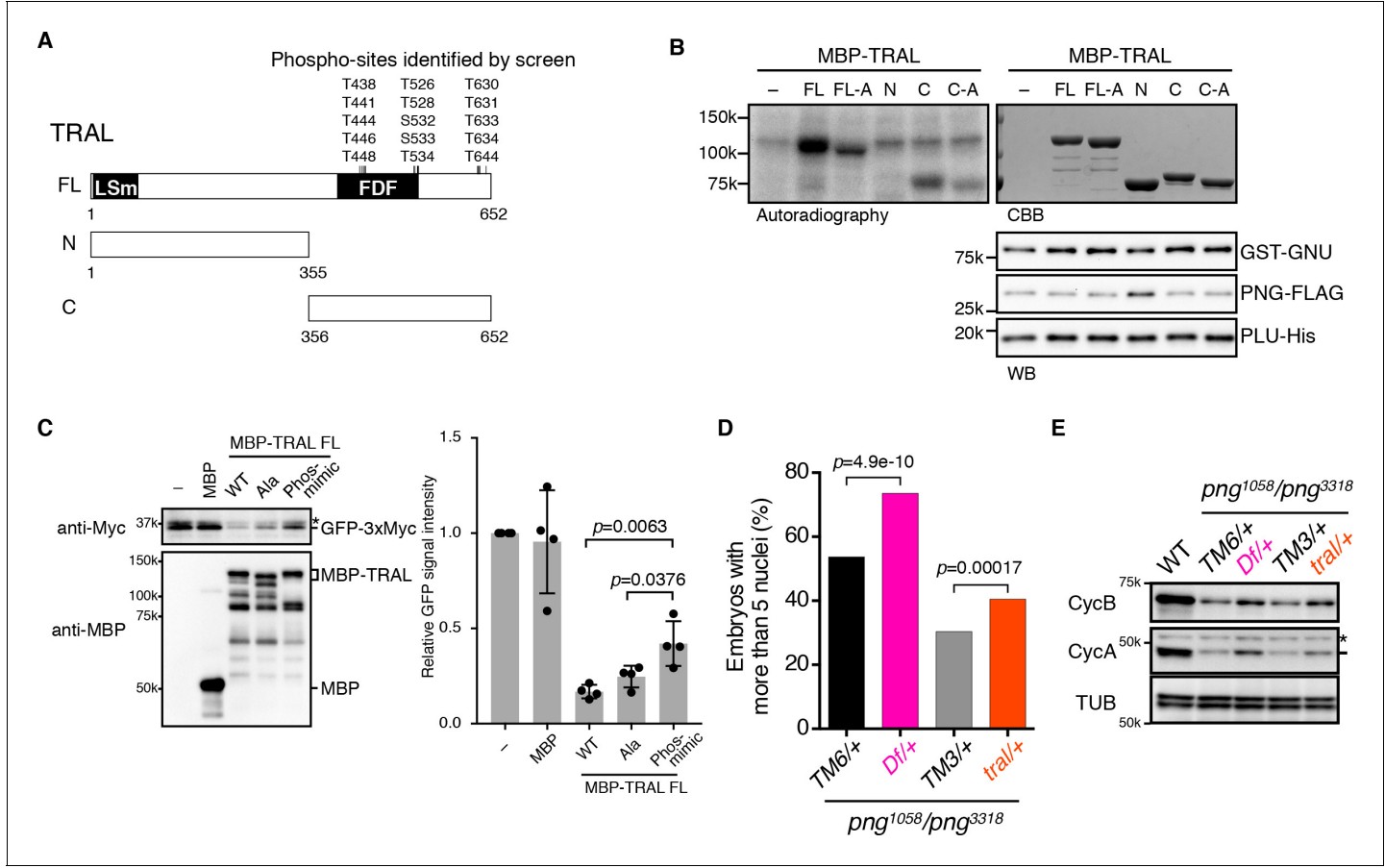

**Figure 3.** PNG kinase directly phosphorylates Trailer hitch (TRAL) and can suppress its repressor function. (A) Schematic representation of Trailer hitch (TRAL) protein. *Drosophila melanogaster* TRAL consists of 652 amino acids and has conserved domains: the LSm (Like Sm) domain is essential for P-body localization of TRAL; the FDF domain binds to ME31B, a translational repressor (*Marnef et al., 2009*). The phosphorylation sites of TRAL identified in the screen map exclusively to the C-terminus. (B) PNG kinase phosphorylates the C-terminus of TRAL in vitro. Maltose-binding protein (MBP)-fused full length (FL) and N- and C-terminal fragments (N: 1–355 and C: 356–652, respectively) of TRAL were expressed and purified from bacteria (A). The MBP-fused TRAL proteins were incubated with the active PNG kinase complex in the presence of radioactive ATP. Incorporated radioactivity in the TRAL proteins was detected by autoradiography. The levels of the MBP-fused TRAL proteins were examined by coomassie staining (CBB). The active PNG kinase complex in the reactions also was examined by immunoblot using anti-GST (GST-GNU), anti-FLAG (PNG-FLAG) and anti-PLU (PLU-His) antibodies. Substitution of the phosphosites in TRAL to alanine (FL-A and C-A) reduced phosphorylation by PNG. The kinase assay was repeated three times. Representative results are shown. (C) Phosphomimetic mutation of the PNG phosphorylation sites in TRAL suppresses translational repression activity of TRAL in vitro. In vitro transcribed *GFP-3xMyc* mRNA was translated in rabbit reticulocyte lysate with or without MBP-fused TRAL wild-type (WT) or MBP-TRAL mutant proteins, which have alanine or aspartic acid in the PNG phosphorylation sites (Ala or Phos-mimic). MBP was used as a control. Translation of *GFP-3xMyc* mRNA was examined by GFP-3xMyc protein levels on an immunoblot using anti-Myc antibody. MBP and MBP-fused TRAL protein levels were examined by immunoblot using anti-MBP antibody. GFP-3xMyc protein levels were quantified and normalized to its levels in the control reaction. Error bars represent standard deviation (n = 4, unpaired t-test; mean ± SD). Representative blots are shown. (D, E) *tral* dominantly suppresses *png* phenotypes. (D) Embryos from females whose genotype were *png1058/png3318* with *Df(3L)ED4483/+* (*Df/+*) or *tral1/+* (*tral/+*) were collected, fixed, and DNA stained with DAPI. Nuclear numbers in the embryos were quantified by fluorescent microscopy. *TM6C, cu1 Sb1/+* (*TM6/+*) and *TM3, Sb1 Ser1/+* (*TM3/+*) were used as controls for *Df/+* and *tral/+*, respectively. The results are the sum of three experiments, and their significance was tested with Fisher's exact test. (E) Cyclin A and B (CycA, CycB) protein levels of the embryos from the females with the indicated genotypes were examined by immunoblot. Alpha-tubulin (TUB) was used as a loading control. The asterisk shows a non-specific band.
DOI: https://doi.org/10.7554/eLife.33150.009

The following source data and figure supplements are available for figure 3:

**Source data 1.** Quantification of GFP protein levels in *Figure 3C*.
DOI: https://doi.org/10.7554/eLife.33150.013
**Source data 2.** Raw data of embryo numbers for *Figure 3D*.
DOI: https://doi.org/10.7554/eLife.33150.014
**Figure supplement 1.** TRAL phosphorylation analysis by quantitative mass spectrometry.
*Figure 3 continued on next page*

*Figure 3 continued*

DOI: https://doi.org/10.7554/eLife.33150.010

**Figure supplement 1—source data 1.** Quantification of phosphopeptide recovery.

DOI: https://doi.org/10.7554/eLife.33150.011

**Figure supplement 2.** Examples of DAPI-stained embryos scored in *Figure 3D*.

DOI: https://doi.org/10.7554/eLife.33150.012

## Interaction between PNG and TRAL in vivo

To confirm that PNG phosphorylates TRAL in vivo we analyzed TRAL phosphorylation by MS following immunoprecipitation from extracts of mature oocytes, in vitro activated oocytes or early embryos. The phosphorylation pattern of TRAL during egg activation was very complex, with many sites. As a consequence, we could find only a small number of phospho-peptides in our quantitative MS analysis, because multiple phosphorylation in a peptide can impede detection of other phospho-sites on the peptide after LC/MS. Nevertheless, we did observe that one of the threonine residues (T644) phosphorylated in vitro became phosphorylated at egg activation in wild-type but not *png* mutant eggs (*Figure 3—figure supplement 1*). Phosphorylation levels of several residues (T35, S59, S472) were reduced in the activated oocytes from the *png* mutant, although they were not found in the substrate screen. These might be potential PNG kinase target sites, but they also could be phosphorylated downstream of PNG indirectly. The proposal of phosphorylation downstream of PNG is consistent with two of these being in the N-terminus of TRAL that is not phosphorylated by PNG in vitro, and the observation that S214 phosphorylation is increased in *png* mutant activated oocytes. Together these results support the conclusion that TRAL is a PNG substrate, but they reveal that TRAL phosphorylation is developmentally dynamic and involves several kinases.

Therefore, we looked for genetic interactions between *png* and *tral* mutants. The *png* gene was identified because mutant females produce eggs that complete meiosis but subsequently fail to initiate mitotic divisions (*Shamanski and Orr-Weaver, 1991*). Nevertheless, DNA replication continues, resulting in embryos with giant, polyploid nuclei. In strong alleles of *png* there is no mitosis, whereas weaker alleles permit a few mitotic divisions but these nuclei ultimately also become polyploid (*Shamanski and Orr-Weaver, 1991*; *Lee et al., 2001*). The absence of mitosis in *png* mutants is due to a failure to promote *cyclin B* mRNA translation at egg activation (*Vardy and Orr-Weaver, 2007*; *Kronja et al., 2014*). We demonstrated that removal of one copy of some genes (such as the translational repressor *pum*, discussed below) can suppress the giant-nuclei *png* phenotype, resulting in embryos that undergo more mitotic divisions and thus have more nuclei (*Lee et al., 2001*). If the gene acts downstream of *png*, this suppression is consistent with *png* acting negatively on the gene. In contrast, removal of one copy of a gene such as *cyclin B* enhances the *png* phenotype, consistent with *png* having a positive effect on this gene (*Lee et al., 2001*).

We compared embryos laid by females with $png^{1058}/png^{3318}$ with one copy of *tral* mutated to sibling controls solely mutant for *png*. Reducing the dosage of *tral* (a heterozygous $tral^1$ mutation, which has a P element insertion) suppressed the *png* phenotype, permitting additional mitoses and increased numbers of nuclei (*Figure 3D*, *Figure 3—figure supplement 2*). This suppression was even more pronounced with a deletion that completely removes the *tral* gene (heterozygous *Df(3L) ED4483*) (*Figure 3D*, *Figure 3—figure supplement 2*). These genetic epistasis results complement the in vitro translation results with the phosphomimetic TRAL form. They are consistent with TRAL being a target of PNG and phosphorylation negatively affecting TRAL.

To test whether the genetic interactions between *tral* and *png* affect *cyclin B* mRNA translation, we examined protein levels by immunoblotting of extracts from the mutant and control embryos. Strikingly, Cyclin B protein levels were increased in the *png* transheterozygous embryos when the dosage of *tral* was reduced (*Figure 3E*). Consistent with the suppression phenotypes, the amount of Cyclin B was restored more with the deletion than with the $tral^1$ allele. Cyclin A, another PNG translational target, also was increased with reduced TRAL.

Taken together, the in vitro and in vivo phosphorylation results and the genetic interaction data indicate that phosphorylation of TRAL by PNG blocks its repressive effects on translation, permitting translation of *cyclin B* at egg activation to permit embryonic mitoses. This could be due to PNG phosphorylation directly repressing TRAL function or via an effect of phosphorylation on the

localization of TRAL. TRAL is present in large cytoplasmic RNP granules in mature oocytes in both Drosophila and *C. elegans*, and these disperse on egg activation (*Weil et al., 2012*; *Noble et al., 2008*). Thus, one model for the effect of PNG on TRAL is that phosphorylation could affect the localization of TRAL to RNP granules. We examined these large visible granules using a *GFP-Tral* FlyTrap line with or without *png* mutations and following TRAL localization during in vitro egg activation. We found that early in activation, by about 10 min, TRAL granules became diminished (*Figure 4A*). In *png* mutant eggs, the TRAL granules also disappeared with normal timing (*Figure 4A*). We conclude that PNG does not appear to be involved in this reorganization of TRAL granules. Indeed, dispersal of TRAL from granules occurs prior to when PNG becomes active at 30 min after egg activation (*Hara et al., 2017*). PNG phosphorylation may more directly affect the ability of TRAL to inhibit translation initiation, as indicated by the effect of the phosphomimetic form on translation in reticulocyte lysates.

## PNG phosphorylates other translational repressors

Given the hundreds of mRNAs whose regulation at egg activation is dependent on PNG, it seemed probable that PNG affects translation through multiple mechanisms and may have multiple substrate targets. We previously showed that the translational repressor *pumilio* (*pum*) dominantly suppresses *png*; a heterozygous mutation of *pum* restores both Cyclin B protein levels and mitosis in *png* mutant embryos (*Vardy and Orr-Weaver, 2007*). Even PUM nonphosphorylated peptides were not recovered in the substrate screen (*Figure 2—source data 1*), therefore, the possibility of PUM being a PNG substrate could not be evaluated. Consequently, we tested for a direct interaction between *png* and *pum* by asking whether PNG can phosphorylate PUM in vitro. A GST-PUM fusion protein is phosphorylated by purified wild-type PNG kinase but not by the kinase-dead form (*Figure 4B*).

The ME31B RNA helicase acts as a translational repressor (*Nakamura et al., 2001*) and is a binding partner to TRAL (*Tritschler et al., 2008*). We did not recover it above the cut off in the substrate screen, although one ME31B phosphopeptide was present in the wild-type but not kinase-dead PNG sample (*Figure 2—source data 1*). Given its interaction with TRAL, we directly tested ME31B in vitro and found that PNG was able to phosphorylate it (*Figure 4C*). Thus, PNG phosphorylation may affect both of these conserved proteins and their role as a complex in controlling translation.

Another translational regulator that is a potential PNG substrate is BICC. BICC binds to the GNU subunit of the PNG complex directly through its SAM domain (*Chicoine et al., 2007*) (Hara and Orr-Weaver unpublished), and BICC also is known to physically interact with TRAL (*Kugler et al., 2009*). We did not, however, recover BICC from the substrate screen. Despite this, PNG readily phosphorylates BICC in vitro (*Figure 4B*).

These results raise the possibility that PNG acts on a number of translational repressors. The two PNG substrate screens likely were not saturating to identify all potential translational repressor targets. The translational repressors Cup and Caprin were recovered in the first substrate screen but not by our criteria in the second. The dominant genetic suppression of *png* observed with mutation of *tral* or *pum* generates the hypothesis that PNG may inactivate multiple translational repressors by phosphorylation to promote translation of different sets of mRNAs at egg activation. It is also possible that PNG's effect on multiple repressors may target a single set of mRNAs localized to RNP granules. For example, ME31B is bound to TRAL. *BicC* genetically interacts with *tral*, the protein appears to localize to the RNP granules in which TRAL and ME31B reside, and it binds to GNU (*Kugler et al., 2009*; *Chicoine et al., 2007*). From these observations, PNG might phosphorylate multiple targets on RNP granules to de-repress translational inhibition of maternal mRNAs at egg activation.

In addition to its effects at egg activation, PNG may indirectly affect translational repressors later in embryogenesis, at a developmental time when PNG appears to be inactivated (*Hara et al., 2017*). In the embryo the TRAL, ME31B, and Cup proteins form an inhibitory complex that represses the translation of maternal mRNAs. These proteins have been shown to be degraded during the maternal-to-zygotic transition, and functional PNG is a prerequisite for this degradation (*Wang et al., 2017*).

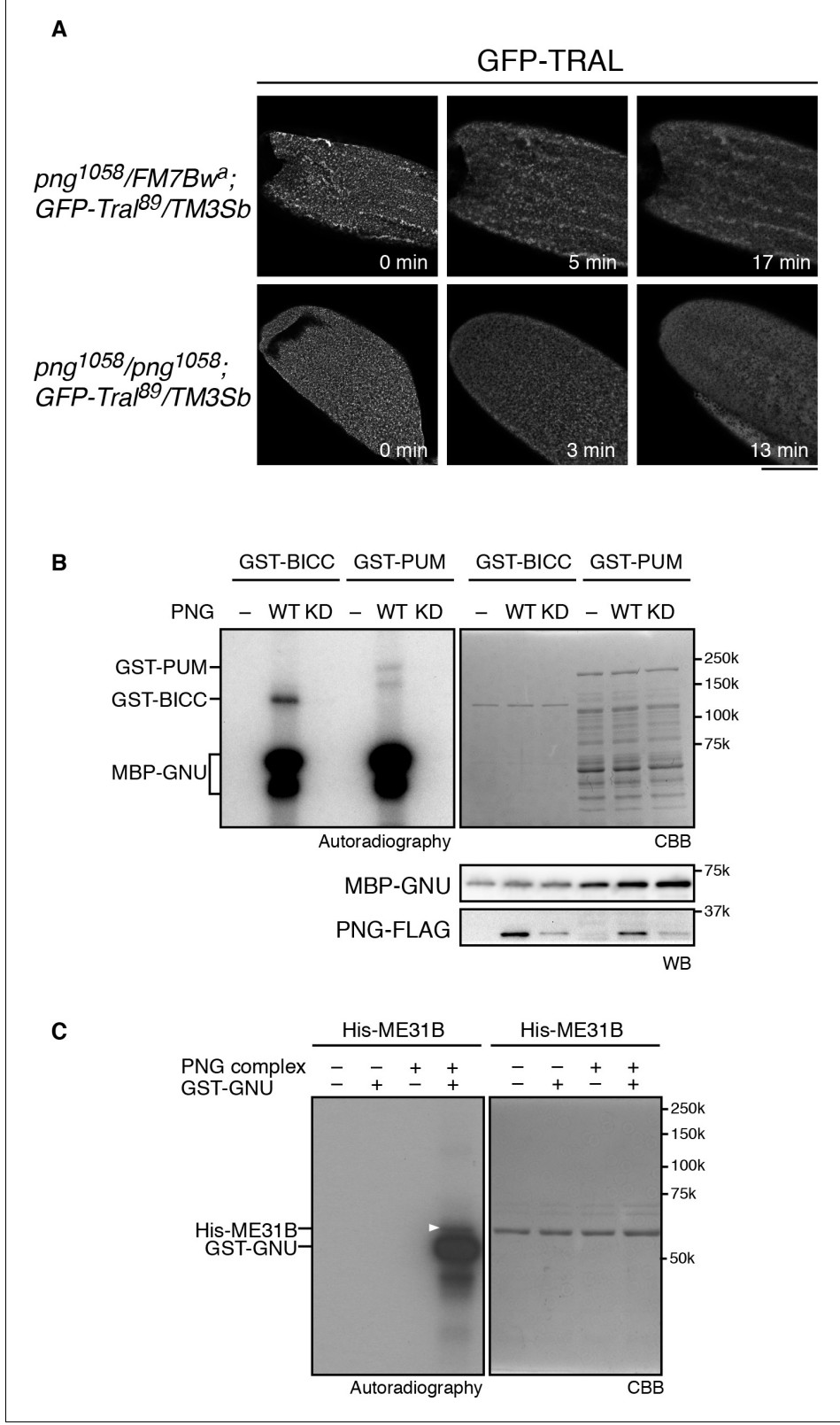

**Figure 4.** PNG kinase phosphorylates translational repressors in vitro. (**A**) TRAL granules in oocytes diminished after egg activation independently of *png*. Heterozygous (*png^1058^/FM7B w^a^*) or homozygous *png* (*png^1058^/png^1058^*) oocytes expressing GFP-Tral (*GFP-Tral^89^/TM3 Sb*) were activated in vitro. GFP-TRAL signal in the oocytes was observed by confocal microscopy. Time in the images indicates time after egg activation. Bar indicates 100 μm.
*Figure 4 continued on next page*

*Figure 4 continued*

Representative oocytes images, taken with the same exposure settings, are shown (WT: n = 8, *png*: n = 3) (B) PNG kinase phosphorylates Bicaudal C (BICC) and Pumilio (PUM). GST-fused BICC and PUM were incubated with or without wild-type (WT) or kinase-dead (KD) PNG kinase activated with MBP-GNU in the presence of radioactive ATP. Radioactivity incorporated into proteins was detected by autoradiography (left panel). The substrate protein levels were examined by coomassie staining (right panel, CBB). MBP-GNU and PNG protein levels were examined by immunoblot using anti-MBP (MBP-GNU) and anti-FLAG (PNG-FLAG) (bottom panels, WB). The kinase assay was repeated in two replicates. Representative results are shown. (C) PNG kinase phosphorylates ME31B in vitro. Recombinant His-ME31B was incubated with or without PNG kinase complex in the presence of radioactive ATP. Because PNG kinase complex requires additional recombinant GNU (GST-GNU) for its full kinase activation, His-ME31B was phosphorylated with or without GST-GNU. Radioactivity incorporated into His-ME31B was detected by autoradiography (left panel). Protein levels of His-ME31B were examined by coomassie staining (right panel, CBB). The white arrowhead indicates phosphorylated His-ME31B protein. The kinase assay was repeated in two replicates. Representative results are shown.

DOI: https://doi.org/10.7554/eLife.33150.015

## Conclusions

We previously showed that the PNG kinase is activated by a signal downstream of egg activation and thus controls massive changes in maternal mRNA translation (*Kronja et al., 2014*; *Hara et al., 2017*). We now have found TRAL is a PNG substrate using a biochemical screen. Phosphorylation by PNG suppressed TRAL's ability to repress mRNA translation. This antagonism also was supported by genetic interaction between *png* and *tral* in fertilized embryos, suggesting that TRAL phosphorylation by PNG during the oocyte-to-embryo transition is a key to remodel maternal mRNAs' translation activity.

The PNG kinase functions as a signal transducer for the external egg activation signal to mRNA translation in the cytoplasm in the activated eggs (*Hara et al., 2017*). Similar strategies can be used in oocyte maturation, during which a hormonal signal leads to phosphorylation of translational regulators to control mRNA translation (*Radford et al., 2008*). In neurons, stimuli cause translocation of mRNA followed by translational activation (*Yoon et al., 2016*). Understanding signaling pathways that transmit extracellular signals to translational controls thus is likely to provide us insight into molecular mechanisms in fertility as well as synaptic plasticity and memory.

## Materials and methods

**Key resources table**

| Reagent type (species) or resource | Designation | Source or reference | Identifiers | Additional information |
|---|---|---|---|---|
| gene (*Drosophila melanogaster*) | *Bicaudal C (BicC)* | NA | FLYB: FBgn0000182 | |
| gene (*D. melanogaster*) | *giant nuclei (gnu)* | NA | FLYB: FBgn0001120 | |
| gene (*D. melanogaster*) | *maternal expression at 31B (me31B)* | NA | FLYB: FBgn0004419 | |
| gene (*D. melanogaster*) | *pan gu (png)* | NA | FLYB: FBgn0000826 | |
| gene (*D. melanogaster*) | *plutonium (plu)* | NA | FLYB: FBgn0003114 | |
| gene (*D. melanogaster*) | *pumilio (pum)* | NA | FLYB: FBgn0003165 | |
| gene (*D. melanogaster*) | *trailer hitch (tral)* | NA | FLYB: FBgn0041775 | |
| strain, strain background (*D. melanogaster*) | WT: *OregonR* | NA | | |

*Continued on next page*

*Continued*

| Reagent type (species) or resource | Designation | Source or reference | Identifiers | Additional information |
|---|---|---|---|---|
| genetic reagent (*D. melanogaster*) | *Df(3L)ED4483* | Bloomington Drosophila Stock Center | BDSC:8070; RRID:BDSC_8070; FLYB:FBst0008070 | FlyBase symbol: *Df(3L)ED4483*; Genotype: *w[1118]; Df(3L)ED4483, P{w[+mW. Scer\FRT.hs3]=3'.RS5+3.3'} ED4483/TM6C, cu[1] Sb[1]* |
| genetic reagent (*D. melanogaster*) | *gfp-tral[89]* | The Flytrap Project; (*Morin et al., 2001*); PMID:11742088 | Flytrap:G00089; DGRC:110584; RRID:DGGR_110584 | Genotype: *w[*]; P{w[+mC]=PTT-un1}G00089* |
| genetic reagent (*D. melanogaster*) | *png[1058]* | (*Shamanski and Orr-Weaver, 1991*); PMID:1913810 | | |
| genetic reagent (*D. melanogaster*) | *png[3318]* | (*Shamanski and Orr-Weaver, 1991*); PMID:1913810 | | |
| genetic reagent (*D. melanogaster*) | *tral[1]* | Bloomington Drosophila Stock Center; (*Wilhelm et al., 2005*); PMID: 16256742 | BDSC:14933; RRID:BDSC_14933; FLYB:FBst0014933 | *Genotype: y[1]; P{y[+mDint2] w[BR.E.BR]= SUPorP}tral[KG08052] ry[506]/TM3, Sb[1] Ser[1]* |
| antibody | alkylated thiophosphate antibody (Rabbit monoclonal) | Abcam | Abcam:ab92570; RRID:AB_10562142 | Anti-Thiophosphate ester antibody [51-8] (1/2000 in 5% Skim milk TBS-T) |
| antibody | anti-PNG (Rabbit polyclonal) | (*Hara et al., 2017*); PMID: 28555567 | | (1/1000 in Hikari solution A) |
| antibody | anti-PLU (Rabbit polyclonal) | (*Elfring et al., 1997*); PMID: 9247640 | | Affinity-purified (1/200 in Hikari solution A) |
| antibody | anti-GNU (Guinea pig polyclonal) | (*Lee et al., 2003*); PMID: 14665672 | | (1/5000 in TBS-T) |
| antibody | anti-TRAL (Rat polyclonal) | (*Tritschler et al., 2008*); PMID:18765641 | | |
| antibody | anti-FLAG (Mouse monoclonal) | Sigma-Aldrich | Sigma-Aldrich:F1804; RRID:AB_262044 | (1/2000 in TBS-T) |
| antibody | anti-MBP (Rat monoclonal) | Sigma-Aldrich | Sigma-Aldrich:SAB4200082 | (1/2000 in Hikari solution A) |
| antibody | anti-Myc (Mouse monoclonal) | Covance | Covance:MMS-150R-1000; RRID:AB_291325 | 9E10; (1/2000 in TBS-T) |
| antibody | anti-GST (Mouse monoclonal) | MBL | MBL:PM013-7; RRID:AB_10598029 | Anti-GST-tag pAb-HRP-DirecT; (1/5000 in Hikari solution A) |
| antibody | HRP-conjugated anti-rabbit IgG | Jackson Immuno Research | Jackson ImmunoResearch: 711-035-152; RRID:AB_10015282 | (1/10000 in TBS-T or Hikari solution B) |
| antibody | HRP-conjugated anti-guinea pig IgG | Jackson Immuno Research | Jackson Immuno Research:706-035-148: RRID:AB_2340447 | (1/50000 in TBS-T) |
| antibody | HRP-conjugated anti-mouse IgG | Jackson Immuno Research | Jackson Immuno Research:115-035-164: RRID:AB_2338510 | (1/20000 in TBS-T) |
| antibody | HRP-conjugated anti-rat IgG | Jackson Immuno Research | Jackson Immuno Research:112-035-062: RRID:AB_2338133 | (1/5000 in TBS-T) |
| antibody | mouse monoclonal anti-Cyclin A | Developmental Studies Hybridoma Bank | DSHB Cat #A12 RRID:AB_528188 | 1/100 in Hikari Solution A |
| antibody | mouse monoclonal anti-Cyclin B | Developmental Studies Hybridoma Bank | DSHB Cat#F2F4 RRID:AB_528189 | 1/200 in Hikari Solution B |

*Continued on next page*

*Continued*

| Reagent type (species) or resource | Designation | Source or reference | Identifiers | Additional information |
|---|---|---|---|---|
| recombinant DNA reagent | pFastBac Dual | Thermo Fisher | Thermo Fisher:10712024 | |
| recombinant DNA reagent | pFastBac1 | Thermo Fisher | Thermo Fihser:10359016 | |
| recombinant DNA reagent | pGEX-6P-1 | GE Healthcare | GE Healthcare:28954648 | |
| recombinant DNA reagent | pMAL-c2X | New England Biolabs | New England Biolabs:N8076S | |
| recombinant DNA reagent | pSP64 Poly(A) | Promega | Promega:P1241 | |
| recombinant DNA reagent | pET28b | Merck | Merck:69865- 3 | |
| recombinant DNA reagent | pFastBac Dual PNG/PLU | (*Hara et al., 2017*); PMID: 28555567 | | |
| recombinant DNA reagent | pFastBac Dual PNG$^{172}$/PLU | This paper | | $png^{172}$:kinase dead mutant (G157E) |
| recombinant DNA reagent | pFastBac1 GNU | (*Lee et al., 2003*); PMID: 14665672 | | |
| recombinant DNA reagent | pFastBac Dual PNG M87G/PLU | This paper | | PNG M87G: gatekeeper mutant |
| recombinant DNA reagent | pFastBac Dual PNG M87A/PLU | This paper | | PNG M87A: gatekeeper mutant |
| recombinant DNA reagent | pGEX-6P-1 GNU | (*Hara et al., 2017*); PMID: 28555567 | | |
| recombinant DNA reagent | pGEX-6P-1 BICC | This paper | | |
| recombinant DNA reagent | pGEX-6P-1 PUMILIO | This paper | | |
| recombinant DNA reagent | pMAL-c2X GNU | This paper | | |
| recombinant DNA reagent | pMAL-c2X TRAL FL | This paper | | |
| recombinant DNA reagent | pMAL-c2X TRAL N | This paper | | TRAL 1–355 |
| recombinant DNA reagent | pMAL-c2X TRAL C | This paper | | TRAL 356–652 |
| recombinant DNA reagent | pMAL-c2X TRAL FL A | This paper | | T438A, T441A, T444A, T446A, T448A, T526A, T528A, S532A, S533A, T534A, T630A, T631A, T633A, T634A, T644A |
| recombinant DNA reagent | pMAL-c2X TRAL C A | This paper | | T438A, T441A, T444A, T446A, T448A, T526A, T528A, S532A, S533A, T534A, T630A, T631A, T633A, T634A, T644A |
| recombinant DNA reagent | pMAL-c2X TRAL FL Phos-mimic | This paper | | T438D, T441D, T444D, T446D, T448D, T526D, T528D, S532D, S533D, T534D, T630D, T631D, T633D, T634D, T644D |
| recombinant DNA reagent | pSP64 Poly(A) EGFP-3xMyc | This paper | | |
| recombinant DNA reagent | pET28b ME31B-3xMyc | This paper | | |

*Continued on next page*

*Continued*

| Reagent type (species) or resource | Designation | Source or reference | Identifiers | Additional information |
|---|---|---|---|---|
| commercial assay or kit | mMESSAGE mMACHINE SP6 Transcription Kit | Thermo Fisher | Thermo Fisher:AM1340 | |
| commercial assay or kit | Rabbit Reticulocyte Lysate System, Nuclease Treated | Promega | Promega: L4960 | |
| commercial assay or kit | TMTsixplex Isobaric Label Reagent Set | Thermo Fisher | Thermo Fisher:90061 | |
| chemical compound, drug | FSBA | SIGMA-Aldrich | SIGMA-Aldrich:F9128 | 5'-(4-fluorosulphonylbenzoyl) adenosine |
| chemical compound, drug | PNBM | Abcam | Abcam:ab138910 | p-Nitrobenzyl mesylate |
| chemical compound, drug | N$^6$(benzyl) ATP-γS | Axxora | Axxora:BLG-B072 | |
| chemical compound, drug | N$^6$(phenethyl) ATP-γS | Axxora | Axxora:BLG-P026 | N$^6$(Phenylethyl) ATP-γ-S |
| chemical compound, drug | N$^6$(furfuryl) ATP-γS | Axxora | Axxora:BLG-F008 | |
| chemical compound, drug | HIKARI signal enhancer | Nacalai | Nacalai:02270–81 | Signal Enhancer HIKARI for Western Blotting and ELISA |
| software, algorithm | WebLogo | (*Crooks et al., 2004*); PMID:15173120 | RRID:SCR_010236 | |
| software, algorithm | Proteome Discoverer | Thermo Fisher | RRID:SCR_014477 | |
| software, algorithm | Mascot | Matrix Science | RRID:SCR_014322 | |
| software, algorithm | CAMV | (*Curran et al., 2013*); PMID:23500044 | | |

## Fly stocks and embryo collection

*Oregon R* was used as the wild-type control. The mutants we used were: *png*[1058] and *png*[3318] (*Shamanski and Orr-Weaver, 1991*); *tral*[1] and *Df(3L)ED4483* (*Wilhelm et al., 2005*) (Bloomington stock center); *GFP-Tral*[89] (*Morin et al., 2001*) (FlyTrap project). Flies were maintained at 22 or 25°C on standard Drosophila cornmeal molasses food.

## Positional scanning peptide libraries

To examine whether PNG kinase had preferred phospho motifs, we screened a positional scanning peptide library as described (*Mok et al., 2010*). Peptide mixtures (50 μM) having the general sequence Y-x-x-x-x-x-S/T-x-x-x-x-A-G-K-K(biotin) were incubated with 1.5 ng/μL of either of the purified recombinant active wild-type or kinase-deficient (PNG[172]) (*Fenger et al., 2000*) PNG kinase (*Hara et al., 2017*) and 4.9 ng/μL GST-GNU at 30 °C for 2 hr in kinase reaction buffer (50 mM Tris-HCl pH 7.5, 10 mM MgCl$_2$, 3 mM MnCl$_2$, 1 mM DTT, 80 mM β-glycerophosphate (βGP), 0.1% BSA, 0.1% Tween 20) containing 50 μM γ[$^{33}$P]-ATP (0.03 μCi/μL). Peptide aliquots were transferred to streptavidin-coated membrane, which was processed as described (*Mok et al., 2010*). Radiolabel incorporation was quantified by phosphor imaging using Quantity One software (Bio Rad). The phosphorylation motifs were visualized using WebLogo (*Crooks et al., 2004*).

## PNG kinase substrate screen

Wild-type embryos (0–2 hr) were collected (*Hara et al., 2017*), homogenized in embryo lysis buffer [50 mM Tris-HCl pH8.0, 150 mM NaCl, 15 NP-40, 1 mM DTT, 2.5 mM EGTA, Complete EDTA-free protease inhibitor (Roche, Indianapolis, IN)], and then sonicated. After centrifugation at 14 krpm for 15 min at 4°C, the supernatant was applied to a PD10 desalting column (GE Healthcare, Waukesha,

WI) to change its buffer into kinase buffer [20 mM Tris-HCl pH7.5, 3 mM $MnCl_2$, 10 mM $MgCl_2$, 80 mM βGP, 0.5mM DTT, Complete EDTA-free protease inhibitor (Roche, Indianapolis, IN)] and treated with 1 mM FSBA for 45 min at room temperature. The extracts were fractionated with ammonium sulfate precipitation at 25%, 40%, 50%, 60% and 75% saturation, and the precipitates of each fraction were frozen in liquid $N_2$ and stored at −80˚C. They were separately resuspended in the same volume of kinase buffer as the initial extract volume and were dialyzed in kinase buffer to remove ammonium sulfate and free FSBA. If fractions had remaining endogenous kinase activities, they were treated with FSBA again and dialyzed to remove it. To thio-phosphorylate proteins in the fractions, 500 µL of each fraction was incubated with the recombinant wild-type or kinase-dead PNG kinase complex (600 ng of PNG-FLAG), GST-GNU (2 µg) to activate the kinase (*Hara et al., 2017*), and 1 mM ATP-γS in kinase buffer supplemented with 0.1 µM okadaic acid at 30˚C for 30 min. Twenty microliters of the reaction were taken from each and alkylated with PNBM to test for thio-phosphorylation by immunoblot using an alkylated thiophosphate antibody (Anti-Thiophosphate ester antibody [51-8]; Abcam. Cambridge, MA) (*Allen et al., 2007*). The remainder of each reaction was methanol-chloroform extracted and trypsinized. In the first experiment, the 25% ammonium sulfate fraction and a pooled fraction (40%, 50%, 60% and 75%) were analyzed, whereas all ammonium sulfate fractions were combined and analyzed in the second experiment. Thio-phosphorylated peptides were captured onto an iodoacetyl resin, stringently rinsed, eluted by oxidation with Oxone, and identified by MS as previously described (*Rothenberg et al., 2016*).

## Recombinant proteins

The regions of the *tral* cDNA corresponding to coding frame for full length (FL: 1–652 amino acids), the N-terminus (1–355 amino acids) or the C-terminus (356–652 amino acids) of the TRAL protein were cloned into pMAL-c2x (NEB, Ipswich, MA) to express maltose binding protein (MBP) fusion proteins in bacteria. Phosphomutants of the residues identified as PNG phosphorylation sites in the substrate screen were made in the *tral* C-terminus cDNA. The threonine or serine phosphosites were substituted to alanine (A) or aspartic acid (Phos-mimic) by gene synthesis (GENEWIZ, South Plainfield, NJ). The mutated cDNAs were cloned into pMAL-c2x (NEB, Ipswich, MA) for analysis of solely the C-terminal fragment or were swapped with the C-terminus of the pMAL-c2x TRAL Full Length (FL) cDNA clone to make pMAP-c2x TRAL FL A or Phos-mimic. MBP-fusion proteins were expressed and purified from bacteria following manufacturer protocols (NEB, Ipswich, MA) and dialyzed in TBS with 0.05% NP-40 and 1 mM DTT.

PUM and BICC FL cDNA were cloned into pGEX-6P-1 (GE Healthcare, Waukesha, WI) to express them as GST fusion proteins in bacteria. The fusion proteins were purified using manufacturer protocols (GE Healthcare, Waukesha, WI) and dialyzed in TBS with 0.05% NP-40 and 1 mM DTT.

To make MBP-fused GNU, GNU cDNA was cloned into pMAL-c2x (NEB, Ipswich, MA), expressed and purified from bacteria as above.

ME31B fused with 3xMyc was cloned into pET28b to express as a His-tagged protein in bacteria. The protein was purified using Ni-NTA beads (Qiagen) and following manufacturer protocols.

## In vitro kinase assay

Two µg of MBP-TRAL were incubated with the recombinant PNG kinase complex (6 ng of PNG-FLAG) and 20 ng GST-GNU to activate PNG kinase at 30˚C for 5 min in 10 µL of kinase buffer2 [20 mM Tris-HCl pH7.5, 3 mM $MnCl_2$, 10 mM $MgCl_2$, 80 mM βGP, 10 µM ATP, Complete EDTA-free protease inhibitor (Roche, Indianapolis, IN)] in the presence of 11.1 MBq/mL [γ-$^{32}$P]ATP. Reactions were terminated by adding 5 µL of 3x Laemmli sample buffer (LSB) with 25 mM EDTA and boiling. Samples were separated on 7.5% SDS-PAGE, and after Coomassie Brilliant Blue (CBB) staining phosphorylated TRAL was detected by autoradiography. The recombinant PNG kinase components were examined by immunoblot as described before (*Hara et al., 2017*). GST-PUM and GST-BICC were treated with WT or kinase-dead PNG kinase complex, and their phosphorylation was detected as above except that MBP-GNU was used to activate PNG kinase instead of GST-GNU.

## In vitro translation assay

*EGFP-3x Myc* cDNA mRNA was in vitro transcribed from its cDNA cloned into pSP64 poly(A) using mMESSAGE mMACHINE (ThermoFisher, Waltham, MA). The mRNA was translated in the rabbit

reticulocyte lysate system (Promega, Madison, WI) with or without 1 µM (final) MBP or MBP-TRAL proteins. Synthesized EGFP-3x Myc protein and MBP or MBP-TRAL proteins from the reaction were examined by immunoblot with anti-Myc (9E10; Covance, Princeton, NJ) or Anti-MBP antibody (Sigma-Aldrich, St. Louis, MO).

## Embryo collection and staining

Fertilized embryos were collected for 2 hr and aged for 1 hr, dechorionated, fixed and the DNA stained with DAPI (*Pesin and Orr-Weaver, 2007*). Their nuclear number was scored as previously described (*Lee et al., 2001*). For immunoblots, dechorionated embryos were lysed in 1x LSB and Cyclin A and B proteins were probed as described (*Hara et al., 2017*).

## In vitro activation and GFP-TRAL imaging

Stage 14 oocytes were collected from *png^1058^/FM7w^a^;GFP-Tral^89^/TM3Sb* or *png^1058^/png^1058^;GFP-Tral^89^/TM3Sb* females in isolation buffer and placed between a glass slide and cover slip separated with double sticky tape as a spacer. The oocytes in the chamber were activated with activation buffer (*Mahowald et al., 1983*). GFP-TRAL in the cytoplasm was inspected by confocal microscopy (Zeiss LSM700).

## Trailer Hitch phospho-site mapping

Stage 14 oocytes, in vitro activated eggs (30 min) and fertilized embryos (1 hr collection) from WT (*OrR*) or *png* mutant (*png^1058/1058^*) females were homogenized in lysis buffer supplemented with RNase A (0.1 mg/mL). Soluble fractions were recovered after centrifugation and their protein concentration was adjusted to 5 µg/µL. Forty µL of the soluble fractions was used for immunoprecipitaion for TRAL. Protein G Dynabeads (Thermo Fisher Scientific, Waltham, MA) were incubated with anti-TRAL antibody (a gift from Izaurralde lab), washed and cross-linked by BS$^3$ (Thermo Fisher Scientific, Waltham, MA) following protocols from the manufacturer. The beads were incubated with the soluble fractions from above for 2 hr at 4°C. After washing the beads three times with lysis buffer, immunoprecipitated proteins were eluted by adding LSB followed by boiling for 5 min. The eluted proteins were run on an SDS-PAGE and stained with CBB.

The TRAL bands were excised from the gel. After destaining with 40% ethanol/10% acetic acid, the proteins were reduced with 20 mM dithiothreitol (Sigma-Aldrich, St. Louis, MO) for 1 hr at 56°C and then alkylated with 60 mM iodoacetamide (Sigma-Aldrich, St. Louis, MO) for 1 hr at 25°C in the dark. Proteins then were digested with 12.5 ng/µL modified trypsin (Promega, Madison, WI) in 50 µL of 100 mM ammonium bicarbonate, pH8.9 at 25°C overnight. Peptides were extracted by incubating the gel pieces with 50% acetonitrile/5% formic acid then 100 mM ammonium bicarbonate, repeated twice followed by incubating the gel pieces with 100% acetonitrile then 100 mM ammonium bicarbonate, repeated twice. Each fraction was collected, combined, and reduced to near dryness in a vacuum centrifuge. Peptides were desalted using C18 SpinTips (Protea, Morgantown, WV) then lyophilized and stored at −80°C.

Peptide labeling with TMT 6plex (Thermo Fisher Scientific, Waltham, MA) was performed per manufacturer's instructions. Samples were dissolved in 70 µL ethanol and 30 µL of 500 mM triethylammonium bicarbonate, pH8.5, and the TMT reagent was dissolved in 30 µL of anhydrous acetonitrile. The solution containing peptides and TMT reagent was vortexed and incubated at room temperature for 1 hr. Samples labeled with the six different isotopic TMT reagents were combined and concentrated to completion in a vacuum centrifuge.

Phosphorylated peptides were enriched as described in (*Ficarro et al., 2009*). In brief, nickel was removed from the Ni-NTA Agarose (Qiagen, Valencia, CA) with 100 mM EDTA. The NTA Agarose was then incubated with 100 mM FeCl$_3$. The peptides were acidified and incubated with the Fe-NTA agarose for 1 hr at room temperature. Phosophopeptides were eluted with 250 mM sodium phosphate.

The peptides were separated by reverse phase HPLC using an EASY- nLC1000 (Thermo Fisher Scientific, Waltham, MA) over a 75 min gradient before nanoelectrospray using a QExactive mass spectrometer (Thermo Fisher Scientific, Waltham, MA). The mass spectrometer was operated in a data-dependent mode. The parameters for the full scan MS were: resolution of 70,000 across 350–2000 $m/z$, AGC 3e$^6$, and maximum IT 50 ms. The full MS scan was followed by MS/MS for the top

10 precursor ions in each cycle with a NCE of 32 and dynamic exclusion of 30 s. Raw mass spectral data files (.raw) were searched using Proteome Discoverer (Thermo Fisher Scientific, Waltham, MA) and Mascot version 2.4.1 (Matrix Science, Boston, MA). Mascot search parameters were: 10 ppm mass tolerance for precursor ions; 10 mmu for fragment ion mass tolerance; two missed cleavages of trypsin; fixed modification were carbamidomethylation of cysteine and TMT 6plex modification of lysines and peptide N-termini; variable modifications were oxidized methionine, serine phosphorylation, threonine phosphorylation, and tyrosine phosphorylation. Only peptides with a Mascot score greater than or equal to 25 and an isolation interference less than or equal to 30 were included in the quantitative data analysis. TMT quantification was obtained using Proteome Discoverer and isotopically corrected per manufacturer's instructions, and the values were normalized to the median of the non-phosphopeptides for each channel. Phosphopeptides were manually validated using CAMV (*Curran et al., 2013*).

## Acknowledgements
We thank Amanda Del Rosario and Eric Spooner for MS analyses and Emir Avilés-Pagán for help with Drosophila stocks. We are grateful to Elisa Izaurralde for the anti-TRAL antibodies. This work was supported by NIH grant R01 GM104047 to BT, by a JSPS Postdoctoral Fellowship for Research Abroad and an Uehara Memorial Foundation Research fellowship to MH, and by NIH grants GM39341 and GM118090 to TO-W. TO-W is an American Cancer Society Research Professor.

## Additional information

### Funding

| Funder | Grant reference number | Author |
|---|---|---|
| Japan Society for the Promotion of Science | | Masatoshi Hara |
| Uehara Memorial Foundation | | Masatoshi Hara |
| National Institutes of Health | GM104047 | Benjamin E Turk |
| National Institutes of Health | GM39341 | Terry L. Orr-Weaver |
| National Institutes of Health | GM118090 | Terry L. Orr-Weaver |

The funders had no role in study design, data collection and interpretation, or the decision to submit the work for publication.

### Author contributions
Masatoshi Hara, Conceptualization, Data curation, Formal analysis, Funding acquisition, Validation, Investigation, Visualization, Methodology, Writing—original draft, Writing—review and editing; Sebastian Lourido, Conceptualization, Data curation, Formal analysis, Validation, Investigation, Visualization, Methodology, Writing—review and editing; Boryana Petrova, Validation, Investigation, Visualization, Methodology, Writing—review and editing; Hua Jane Lou, Validation, Investigation; Jessica R Von Stetina, Investigation, Writing—review and editing; Helena Kashevsky, Validation, Investigation, Methodology; Benjamin E Turk, Supervision, Funding acquisition, Validation, Investigation, Visualization, Methodology, Writing—review and editing; Terry L Orr-Weaver, Conceptualization, Formal analysis, Supervision, Funding acquisition, Validation, Investigation, Visualization, Methodology, Writing—original draft, Project administration, Writing—review and editing

### Author ORCIDs
Masatoshi Hara http://orcid.org/0000-0001-8433-1111
Sebastian Lourido http://orcid.org/0000-0002-5237-1095
Terry L Orr-Weaver http://orcid.org/0000-0002-7934-111X

### Decision letter and Author response
Decision letter https://doi.org/10.7554/eLife.33150.019

Author response https://doi.org/10.7554/eLife.33150.020

---

## Additional files

### Supplementary files
• Transparent reporting form
DOI: https://doi.org/10.7554/eLife.33150.016

---

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
