## [Decision Letter]

Thank you for submitting your article "Identification of PNG kinase substrates uncovers interactions with the translational repressor Tral in the oocyte-to-embryo transition" for consideration by *eLife*. Your article has been reviewed by two peer reviewers, and the evaluation has been overseen by a Reviewing Editor and Kevin Struhl as the Senior Editor. The reviewers have opted to remain anonymous.

The reviewers have discussed the reviews with one another and the Reviewing Editor has drafted this decision to help you prepare a revised submission.

Summary:

Hara et al. follow up their *eLife* paper on the control of *Drosophila* PNG paper during egg activation to show that PNG is an active kinase and identify a major target as Tral, a translational repressor. They find that phosphorylation of Tral likely inhibits its repression of translation of proteins such as cyclins A and B, and that its action on their expression is within a pathway with PNG. These data provide a mechanistic explanation for how PNG activates translation of SMG, which then regulates translation of certain mRNAs. The authors further propose that PNG might regulate other translational regulators and examined three: PLU, Bic-C, and Me31B. They show that all three can be phosphorylated by PNG in vitro, although they do not test whether this affects their activity. Such a test would have strengthened the paper further but would require a separate large study, and the Tral data already support the idea. The authors hypothesize that their data on Tral show, and on other proteins suggest, that PNG phosphorylation inactivates translational repressors thereby activating translation upon egg activation.

Essential revisions:

No additional experiments are required but the following comments should be addressed by re-writing the manuscript and including extant raw data:

1) Figure 2: the authors performed two replicates of MS to identify potential substrates of PNG, but only report detailed results for one replicate. The complete data from both replicates should be presented. In Figure 2—source data 1, the spectral count column headings should read "peptide" rather than "protein." Finally, the authors should include an additional supplemental spreadsheet that lists the actual peptide sequences obtained from this analysis.

Related to this, Figure 2—figure supplement 3 shows the overlap between two experiments to define PNG targets. It was of concern that only about 1/3 of the hits were found in both runs. Some consideration or explanation for this would be important. Perhaps this also relates to the failure to detect PUM and BIC-C in the target datasets. Being conservative in focusing on the overlap genes is valid, but this raises a question about the sensitivity of the screen. Please discuss.

2) PNG itself is one of its targets. It would be interesting to investigate or at least speculate on this further. Could it be inactivating itself, or alternatively activating itself?

3) Subsection “Identification of PNG substrates”, last sentence: is there any experimental basis for the statement that PLU or GNU may provide specificity for targets of PNG phosphorylation, or is this just speculation?

Subsection “PNG phosphorylates other translational repressors”, last paragraph: what is the basis for the statement that phosphorylation of other translational repressors inactivate them to promote translation? Other targets may be regulated differently by phosphorylation.

Figure 2: the row containing RpL40 is improperly indented, making it misaligned with the rest of the table.

Figure 3: while mutation of the predicted PNG phosphorylation sites in TRAL certainly appears to reduce levels of phosphorylation, contrary to what the authors state in the corresponding figure legend, phosphorylation is not reduced to background. This therefore implies that there are other sites in the C-terminus of TRAL that are phosphorylated by PNG. For example, FL-A and C-A mutations of MBP-TRAL retain some level of phosphorylation, suggesting that other sites of phosphorylation by PNG have either not been detected by mass spec or, if detected, were not mutated. This concern is supported in Figure 3—figure – supplement 1, where several in vivo phosphorylation sites are PNG-dependent, but were not included as part of the mass spec results or mutagenesis assay (e.g., T35 and S59 phosphorylation is lost to the same degree as T644 in the *png/png* mutant compared to wild-type). These qualifications and caveats need to be included in the revised manuscript.

Figure 3: Phos-mimic TRAL only slightly (though significantly) decreased translational repression of the reporter by TRAL. Since is clear that the in vitro phosphorylation assay did not capture all of the phosphorylation sites, in order to dissect the extent to which phosphorylation of TRAL is indeed regulating its translational repression activity, it would be informative to combine mutation of all of the sites on TRAL that are phosphorylated in a PNG-dependent manner (from both assays). Additional discussion of this matter must be included.

Figure 3: it is not made clear which alleles are the "*tral* insertion" (I assume "*tral*") vs. "*tral* deletion" (I assume "Df") referred to in the main text. The authors should be more explicit in discussing the nature of the *tral* alleles. Similarly, the discussion in the main text of the experiment in Figure 4 does not specifically describe the genotypes used; this should be clarified.

Figure 3 and Figure 4: what are the asterisks in in the labels for the blots?

Figure 3—figure supplement 1: what are the y-axes in these graphs?

Figure 4: the top lane labels on the left are misaligned.

Subsection “Interaction between PNG and Tral in vivo”, second paragraph and subsection “PNG phosphorylates other translational repressors”, first paragraph: the use of the word "antagonistically" is confusing when discussing epistasis with regards to the PNG phenotype. Saying that a gene acts antagonistically to, or together with *png* suggests that these are regulators or binding partners of PNG protein. However, the genes addressed here are in fact downstream, and regulated either negatively or positively by PNG. For clarity, these sentences should be rewritten. For example: "…removal of one copy of some genes can suppress the giant-nuclei phenotype, while removal of other genes such as cyclin B worsens the phenotype." and: "Thus TRAL suppresses the *png* phenotype, suggesting PNG negatively regulates TRAL function…" similar when discussing *pum* in the aforementioned paragraph.

---

## [Author Response]

Essential revisions:No additional experiments are required but the following comments should be addressed by re-writing the manuscript and including extant raw data:1) Figure 2: the authors performed two replicates of MS to identify potential substrates of PNG, but only report detailed results for one replicate. The complete data from both replicates should be presented. In Figure 2—source data 1, the spectral count column headings should read "peptide" rather than "protein." Finally, the authors should include an additional supplemental spreadsheet that lists the actual peptide sequences obtained from this analysis.Related to this, Figure 2—figure supplement 3 shows the overlap between two experiments to define PNG targets. It was of concern that only about 1/3 of the hits were found in both runs. Some consideration or explanation for this would be important. Perhaps this also relates to the failure to detect PUM and BIC-C in the target datasets. Being conservative in focusing on the overlap genes is valid, but this raises a question about the sensitivity of the screen. Please discuss.

The two mass spec substrate screens were not identical. The first experiment was done as a pilot. In this experiment only wild-type PNG kinase was used, and the total peptide numbers (those that flowed through the iodoacetyl resin and thus were not phophosphorylated) were not recovered and quantified. Additionally, in the first experiment the two ammonium sulfate fractions of 25% and 40-75% were analyzed separately.

The second mass spec substrate screen was a thorough screen, and it was this one that was reported in the submitted manuscript. In this screen wild-type and kinase-dead samples were analyzed in parallel. The peptides that flowed through the iodoacetyl resin were recovered and quantified. The peptides from the 5 ammonium sulfate fractions were processed independently but they were pooled for the mass spec analysis.

The differences between the pilot and second experiment likely explain the restricted overlap between the two experiments. But we also do not think that the two substrate screens were saturating and state this in the revised text.

In response to the reviewers’ comments we reanalyzed the spectra and the data, demanding more restrictive criteria of comparing both experiments by Mascot analysis and removing any substrates for which phosphopeptides also were recovered in the kinase-dead sample. We also made the stringent requirement to only consider as substrates proteins for which two independent phosphopeptides were recovered. This altered the substrates identified in experiment 2 and changed the overlap. The revised Figure 2 presents 11 of the 36 the substrates identified with these more stringent criteria.

We now provide an Excel file (Figure 2—source data 1) with the data from both sets of experiments, including a spreadsheet with the peptide sequences. The revised list comparing the substrates uniquely recovered in experiment 1 or 2 and those recovered in both is present also in this Excel file.

2) PNG itself is one of its targets. It would be interesting to investigate or at least speculate on this further. Could it be inactivating itself, or alternatively activating itself?

With the new analysis, PNG is not identified as a substrate.

3) Subsection “Identification of PNG substrates”, last sentence: is there any experimental basis for the statement that PLU or GNU may provide specificity for targets of PNG phosphorylation, or is this just speculation?

It was just speculation that PLU or GNU could provide substrate specificity. The sentence has been revised to make it clear that this is merely a proposal.

Subsection “PNG phosphorylates other translational repressors”, last paragraph: what is the basis for the statement that phosphorylation of other translational repressors inactivate them to promote translation? Other targets may be regulated differently by phosphorylation.

We previously published genetic interaction data between PNG and PUM that were consistent with PNG inactivating PUM to restore translation of *cyclin B* mRNA (Vardy and Orr-Weaver, 2007). This has been revised to make it clear that the genetic results with *pum* and *tral* are the basis for the hypothesis that PNG phosphorylation of other translational repressors may inactivate them.

Figure 2: the row containing RpL40 is improperly indented, making it misaligned with the rest of the table.

All the rows are now aligned.

Figure 3: while mutation of the predicted PNG phosphorylation sites in TRAL certainly appears to reduce levels of phosphorylation, contrary to what the authors state in the corresponding figure legend, phosphorylation is not reduced to background. This therefore implies that there are other sites in the C-terminus of TRAL that are phosphorylated by PNG. For example, FL-A and C-A mutations of MBP-TRAL retain some level of phosphorylation, suggesting that other sites of phosphorylation by PNG have either not been detected by mass spec or, if detected, were not mutated. This concern is supported in Figure 3—figure supplement 1, where several in vivo phosphorylation sites are PNG-dependent, but were not included as part of the mass spec results or mutagenesis assay (e.g., T35 and S59 phosphorylation is lost to the same degree as T644 in the png/png mutant compared to wild-type). These qualifications and caveats need to be included in the revised manuscript.

We removed the statement from the figure legend that substitution of the phosphosites in TRAL to alanine reduced phosphorylation by PNG to background levels. In the text we now state that there could be other PNG phosphorylation sites in the C-terminus of TRAL that were not detected in the screen.

We have revised the text to state:

“Phosphorylation levels of several residues (T35, S59, S472) were reduced in the activated oocytes from the *png* mutant, although they were not found in the substrate screen. […] The proposal of phosphorylation downstream of PNG is consistent with two of these being in the N-terminus of TRAL that is not phosphorylated by PNG in vitro, and the observation that S214 phosphorylation is increased in *png* mutant activated oocytes.”

Figure 3: Phos-mimic TRAL only slightly (though significantly) decreased translational repression of the reporter by TRAL. Since is clear that the in vitro phosphorylation assay did not capture all of the phosphorylation sites, in order to dissect the extent to which phosphorylation of TRAL is indeed regulating its translational repression activity, it would be informative to combine mutation of all of the sites on TRAL that are phosphorylated in a PNG-dependent manner (from both assays). Additional discussion of this matter must be included.

We now state that “The potential existence of additional PNG phosphorylation sites in the C-terminus of TRAL could account for why suppression of translational repression by the phosphomimetic form of TRAL was not complete”.

Figure 3: it is not made clear which alleles are the "tral insertion" (I assume "tral") vs. "tral deletion" (I assume "Df") referred to in the main text. The authors should be more explicit in discussing the nature of the tral alleles. Similarly, the discussion in the main text of the experiment in Figure 4 does not specifically describe the genotypes used; this should be clarified.

The discussion of the genotypes in the text has been clarified.

Figure 3 and Figure 4: what are the asterisks in in the labels for the blots?

The asterisk in Figure 4 has been removed. In Figure 3 this asterisk shows a non-specific band. This is now mentioned in the Figure 3 legend.

Figure 3—figure supplement 1: what are the y-axes in these graphs?

The Y-axes show the relative value of phosphopeptide abundance. This is now explained in the figure legend.

Figure 4: the top lane labels on the left are misaligned.

This has been corrected.

Subsection “Interaction between PNG and Tral in vivo”, second paragraph and subsection “PNG phosphorylates other translational repressors”, first paragraph: the use of the word "antagonistically" is confusing when discussing epistasis with regards to the PNG phenotype. Saying that a gene acts antagonistically to, or together with png suggests that these are regulators or binding partners of PNG protein. However, the genes addressed here are in fact downstream, and regulated either negatively or positively by PNG. For clarity, these sentences should be rewritten. For example: "…removal of one copy of some genes can suppress the giant-nuclei phenotype, while removal of other genes such as cyclin B worsens the phenotype." and: "Thus TRAL suppresses the png phenotype, suggesting PNG negatively regulates TRAL function…" similar when discussing pum in the aforementioned paragraph.

The text has been revised as suggested to make the genetic interactions clearer.